# Biological Role and Clinical Implications of *MYOD1^L122R^* Mutation in Rhabdomyosarcoma

**DOI:** 10.3390/cancers15061644

**Published:** 2023-03-07

**Authors:** Daniela Di Carlo, Julia Chisholm, Anna Kelsey, Rita Alaggio, Gianni Bisogno, Veronique Minard-Colin, Meriel Jenney, Raquel Dávila Fajardo, Johannes H. M. Merks, Janet M. Shipley, Joanna L. Selfe

**Affiliations:** 1Department of Women’s and Children’s Health, University of Padova, 35128 Padua, Italy; 2Pediatric Hematology-Oncology Division, University Hospital of Padova, 35128 Padova, Italy; 3Children and Young People’s Unit, Royal Marsden Hospital, Institute of Cancer Research, Sutton SM2 5NG, UK; 4Department of Pediatric Histopathology, Manchester University Foundation Trust, Manchester M13 9WL, UK; 5Pathology Unit, Department of Laboratories, Bambino Gesù Children’s Hospital, IRCCS, 00165 Rome, Italy; 6Department of Pediatric and Adolescent Oncology, INSERM U1015, Gustave Roussy, Université Paris-Saclay, 94800 Villejuif, France; 7Department of Pediatric Oncology, Children’s Hospital for Wales, Cardiff CF14 4XW, UK; 8Department of Radiation Oncology, University Medical Center Utrecht, 3584 CX Utrecht, The Netherlands; 9Princess Máxima Center for Pediatric Oncology, 3584 CS Utrecht, The Netherlands; 10Division of Imaging and Oncology, University Medical Center Utrecht, Utrecht University, 3584 CX Utrecht, The Netherlands; 11Sarcoma Molecular Pathology Team, Divisions of Molecular Pathology and Cancer Therapeutics, The Institute of Cancer Research, London SM2 5NG, UK

**Keywords:** *MYOD1*, spindle cell, sclerosing rhabdomyosarcoma, high-risk, children

## Abstract

**Simple Summary:**

The myogenic differentiation 1 gene (*MYOD1*) p.L122R mutation was first discovered in a subset of clinically aggressive rhabdomyosarcomas in both adults and children. It occurs most frequently in spindle cell (Sp) or sclerosing (Sc) RMS, but also occasionally in ERMS, including 3% of all PAX fusion-negative RMS. The presence of the *MYOD1^L122R^* mutation seems to be associated with a very poor prognosis, especially when it occurs concomitantly with other mutations such as *PIK3CA*. In this review, we consider the known biological effects of *MYOD1* mutations and present a review of published cases of RMS with *MYOD1* mutations. Together, the reviewed biological characteristics and the clinical features focus attention on this specific subgroup of patients with poor outcome and highlight the need to identify an optimal therapeutic strategy.

**Abstract:**

Major progress in recent decades has furthered our clinical and biological understanding of rhabdomyosarcoma (RMS) with improved stratification for treatment based on risk factors. Clinical risk factors alone were used to stratify patients for treatment in the European Pediatric Soft Tissue Sarcoma Study Group (EpSSG) RMS 2005 protocol. The current EpSSG overarching study for children and adults with frontline and relapsed rhabdomyosarcoma (FaR-RMS NCT04625907) includes *FOXO1* fusion gene status in place of histology as a risk factor. Additional molecular features of significance have recently been recognized, including the *MYOD1^L122R^* gene mutation. Here, we review biological information showing that *MYOD1^L122R^* blocks cell differentiation and has a MYC-like activity that enhances tumorigenesis and is linked to an aggressive cellular phenotype. *MYOD1^L122R^* mutations can be found together with mutations in other genes, such as *PIK3CA*, as potentially cooperating events. Using Preferred Reporting Items for Systematic Reviews and Meta-Analyses (PRISMA) guidelines, ten publications in the clinical literature involving 72 cases were reviewed. *MYOD1^L122R^* mutation in RMS can occur in both adults and children and is frequent in sclerosing/spindle cell histology, although it is also significantly reported in a subset of embryonal RMS. *MYOD1^L122R^* mutated tumors most frequently arise in the head and neck and extremities and are associated with poor outcome, raising the issue of how to use *MYOD1^L122R^* in risk stratification and how to treat these patients most effectively.

## 1. Introduction

Rhabdomyosarcoma (RMS) is the most frequent soft tissue sarcoma in children and adolescents, accounting for 5% of all pediatric cancers [1]. Major progress has been made in recent decades to improve knowledge of the disease and its biology, with subsequent better stratification of treatment, based on risk factors. Accordingly, survival in recent years has improved, especially for patients with localized tumors, reaching a 5-year overall survival (OS) of 85% for high-risk disease [2]. This has been possible due to risk-adapted, multidisciplinary treatment, with the most recent advance being the introduction of maintenance treatment for high-risk patients [2]. Despite all these efforts, relapse still occurs in 24% of patients [3,4,5,6] and, additionally, RMS is refractory to frontline treatment in a small number of patients, including 2% of intermediate-risk patients [7,8].

Various clinical risk factors were used to stratify treatment in the European Pediatric Sarcoma Study Group (EpSSG) RMS 2005 protocol including histology, primary disease site, nodal involvement, age < or ≥10 years, size ≤5 or >5 cm and distant metastasis [7,9,10]. Moreover, since the introduction of the RMS 2005 protocol, alveolar histology has been better characterized as being associated with the presence of *PAX3::FOXO1* or *PAX7::FOXO1* fusion in 85–90% of alveolar tumors [11]. In addition, fusion status has been found to correlate better than histology with prognosis [12]. Thanks to the progress and the routine use of techniques of molecular characterization, new genetic abnormalities have been progressively identified in RMS, such as new fusion transcripts including *PAX3::NCOA1*, *PAX3::NCOA2*, *VGLL2::NCOA2*, *FUS::TFCP2*, *EWSR::TFCP2* [13,14,15,16], as well as new somatic pathogenic mutations such as *MYOD1* and *TP53* mutations [17,18].

The discovery of myogenic differentiation 1 (*MYOD1*) mutations is closely related to the emerging evidence of spindle cell (SpRMS) and sclerosing (ScRMS) subtypes of RMS as separate entities from the other histologic subtypes. SpRMS was first described in children in a report by Cavazzana et al. in 1992 [19], followed by the sclerosing subtype (ScRMS), described between 2000 and 2004 by different authors, first in adult case series [20,21], then in the pediatric setting [22], with a variable outcome. While in previous classifications Sp/ScRMS was considered a subtype of ERMS, in 2013 Sp/ScRMS subtypes were defined as a stand-alone entity in the WHO classification [23]. Later on, in the 2020 WHO classification, Sp/ScRMS evolved further as it included three different groups: infantile subgroup with recurrent *VGLL2::NCOA2* and other fusions; *MYOD1* mutated subgroup; Sp and epithelioid RMS with *TFCP* fusions. However, spindle RMS outside this context is probably part of the ERMS group [24].

Since the first report from Memorial Sloan Kettering Center published in Nature Genetics in 2014 [25], the *MYOD1^L122R^* somatic mutation has been reported repeatedly in case series and case reports of adult and pediatric patients with Sp/ScRMS [17]. The biology of *MYOD1* and the clinical features of the patients with *MYOD1^L122R^* mutated tumors have not been comprehensively characterized and little is known about its true prognostic role.

The aim of this review is to draw together understanding of the biology of *MYOD1^L122R^* mutation and its frequency in tumors from patients affected by RMS and to explore clinical features of such patients, the prognostic role of *MYOD1^L122R^* mutation and the possible implications for risk stratification and treatment.

Firstly, we reviewed the literature on *MYOD1* biology to give an overview of the normal functioning of the protein and its altered mechanism of action when mutated in RMS to *MYOD1^L122R^*. Secondly, we focused on clinical aspects of patients with *MYOD1^L122R^* mutated RMS through a review of the clinical records reported in the literature. A search in PubMed identified case reports, cases series, reviews and articles including cases of patients (adults and children) with RMS bearing *MYOD1^L122R^* somatic mutations, from the first report of *MYOD1^L122R^* mutation published in 2014 to September 2022, according to PRISMA guidelines (Appendix A). To enlarge the search, we also looked for papers focused on Sp- and ScRMS. Used terms and words were: *MYOD1*, RMS, spindle cell RMS, sclerosing RMS, pediatric RMS, adult RMS. Contributions in English, French and Italian were considered. The minimum data required to include a case record in our analysis were age at diagnosis, sex, site of the primary tumor, histology and confirmation of *MYOD1^L122R^* mutation. Description of events and outcome criteria were looked for, but, since they were not described in detail for most of the cases, they were not mandatory to include the reference in our analysis. Papers with insufficient data for the purpose of our study were excluded. All the cases that were reported twice (or more) through the different reports have been carefully checked for, identified and presented just once. Events and outcome were taken into account only for patients with a minimal follow-up of 12 months.

## 2. Normal Role for MYOD1 in Myogenesis

Myogenic determination factor 1 (MYOD1) is one of the four originally described myogenic regulatory factors identified towards the end of the 1980s [26] (along with myogenin MYF4, MYF5 and MYF6 (also known as MRF4 or herculin) [27]). All of these proteins share the ability to convert non-myogenic cells such as fibroblasts to the muscle lineage upon transfection of their encoding cDNA alone. These four factors are highly conserved in vertebrates, including birds and amphibians [28]. They are expressed sequentially in embryogenesis beginning with *MYF5* then *MYOD1*, *MYF6* and *MYF4*. *MYF5* and *MYOD1* are responsible for the specification and determination of skeletal myoblasts, whereas *MYF4* plays a role in the formation of myofibers (*MYF4* null mice have normal numbers of myoblasts but lack muscle fibers) [29]. Expression of *MYF5* and *MYOD1* protein is absent in quiescent satellite stem cells [30,31,32]. Evidence from knockout mouse models suggests that there is a degree of functional redundancy between *MYOD1* and *MYF5,* as single knockout models of either gene result in normal muscle development. Notably in *MYOD1* null mice, *MYF5* is significantly upregulated as a compensatory mechanism [33,34]. Double knockout mice for *MYOD1* and *MYF5*, in contrast, are born immobile soon after birth with complete skeletal muscle aplasia [34]. Subsequently, it has been shown that attempting to rescue muscle formation in these mice by replacing the *MYOD1* coding sequence with that of *MYF5* is unsuccessful, showing that temporal and spatial expressions of these genes are distinct [35]. Additional evidence that these two genes cannot fully compensate for each other in all contexts comes from experimental models demonstrating that *MYOD1* deficient mice were unable to regenerate muscle following injury due to myogenic precursor cells continuing to self-renew rather than enter the differentiation pathway following satellite cell activation [36].

The four primary myogenic factors also share homology at the proteomic level with a very large transcription factor superfamily, the basic helix–loop–helix (bHLH) family, which comprises hundreds of members. These proteins bind E-box motifs in the genome (CANNTG) following dimerization; in the case of MYOD1, heterodimerization occurs with a member of the E-protein bHLH family [37]. The MYOD1:E-protein heterodimer shows a preference for binding the VCASCTGT E-box site (where V indicates not T, and S indicates G or C). Transcriptional regulation of myogenesis by MYOD1 is multifaceted and complex, as E-box binding motifs occur very frequently in the genome, allowing MYOD1 to bind at thousands of different locations. The majority of loci bound by MYOD1 are therefore not muscle-specific genes and indeed MYOD1 has been found to affect regional transcription by causing widespread histone acetylation through recruiting histone acetyltransferases (HATs) and opening up chromatin to other transcription factors and machinery [38]. This indirect mode of regulation means that binding of MYOD1 to a gene promoter may not result in any direct expression of that gene. MYOD1 primarily drives progression of muscle differentiation by direct transactivation, via use of “feed-forward” circuits whereby MYOD1 activates transcription of muscle-specific transcription factors which later in the differentiation pathway activate a further set of genes by co-operatively binding with MYOD1 [39]. MYOD1 also participates in negative (termed incoherent) feed-forward circuits to achieve temporal co-ordination of myogenic differentiation where MYOD1 activates proteins that subsequently compete with it for promoter or enhancer binding at muscle gene loci [40].

## 3. *MYOD1^L122R^* Mutation Blocks Differentiation of the Cells and Has an MYC-like Activity

A very early study by Van Antwerp et al., published long before any *MYOD1* mutations had been found in RMS, investigated the consequences of mutating various residues in the basic DNA-binding domain of MYOD1 to the analogous residue of MYC, the oncoprotein which is also a member of the bHLH transcription factor family [41]. This study identified one mutation which allowed MYOD1 to continue to bind MYOD1 binding sites but no longer transactivated target genes, and additionally was now also capable of binding MYC binding sites and repressing a reporter construct phenocopying the action of MYC. This mutation was the substitution of leucine with arginine (L122R), the recurrent mutation observed in RMS. The authors prophetically predicted that this mutation (achieved by a single base substitution in the coding sequence of *MYOD1* DNA) could have a role in skeletal muscle oncogenesis. Following the first detection of the L122R mutation in RMS tumors, further functional work confirmed these initial findings. Kohsaka et al. introduced ectopic expression of *MYOD1^L122R^* into C2C12 mouse myoblasts and demonstrated that they were no longer able to be induced to differentiate and express muscle markers or undergo fusion to form myofibers in comparison to wild-type MYOD1 or controls [25]. This effect is seen in the presence of the normal wild-type MYOD1, suggesting that MYOD1^L122R^ can act in a dominant negative manner. Gene expression profiling of the mouse myoblast models supported a shift in expression pattern towards a more *MYC*-like signature and, similarly, analysis of histone modification by ChIP sequencing showed a stronger deposition of activating histone marks (H3K4me3) at MYC target genes compared to wild-type MYOD1. The overall picture suggests that the binding site repertoire of mutant MYOD1 has expanded to include *MYC*-regulated genes while still retaining the ability to bind MYOD1 sites at myogenic gene loci, yet blocking expression of these genes by competing with normal MYOD1 and preventing differentiation. In comparison with normal myoblasts, RMS typically expresses both MYOD1 and myogenin yet fails to undergo terminal differentiation. MYOD1 was found to bind myogenic loci more poorly in the RD embryonal RMS cell line compared to primary human myoblasts, therefore it is unclear what the contribution of *MYOD1^L122R^* would be to an existing underlying defect in differentiation in embryonal RMS [42]. A schema for the functional behavior of MYOD1^L122R^ is summarized in Figure 1.

## 4. *MYOD1^L122R^* Affects Tumorigenesis and Causes an Aggressive Biological Phenotype

Evidence of the phenotypic behavior of cells harboring the *MYOD1^L122R^* mutation from the mouse myoblast model generated by Kohsaka et al. indicates more aggressive tumorigenic behavior [25]. *MYOD1^L122R^*-expressing cells do not proliferate any faster than controls but do form more colonies in the anchorage-independent soft agar assay (a tumorigenic property that predicts tumor formation in nude mice). C2C12 myoblast cells expressing exogenous *MYOD1^L122R^* exhibited tumor formation in in vivo xenograft models in comparison to the parental cell line which does not form tumors in mice, however, when the *MYOD1^L122R^* mutation was combined with an activating mutation in *PIK3CA*, a dramatic increase in tumor size was observed.

## 5. *MYOD1^L122R^* Mutation Frequently Occurs with Other Mutations

A large panel sequencing study of somatic mutations in RMS tumor samples, which identified 17 *MYOD1^L122R^* cases, illustrates the enrichment of other mutations found in mutant *MYOD1* cohorts, particularly *PIK3CA* and other *PI3K* pathway genes which are involved in approximately half of mutant *MYOD1* tumors, and the *RAS* pathway which accounts for at least another quarter [18]. Mutations in these pathways are not mutually exclusive, with some tumors containing *RAS*, *PIK3CA* and *MYOD1* mutations. This suggests that *MYOD1* may require these cooperating mutations to achieve its full spectrum of tumorigenic behavior. Interestingly, in the Kohsaka et al. study, an activating mutation in *PIK3CA* led to increased proliferation but not increased soft agar colony formation and therefore the mutations in *MYOD1* and *PIK3CA* led to complementary but not overlapping phenotypes in mouse myoblasts [25]. Further understanding of the possible cooperative nature and dependencies on the different mutations associated with *MYOD1^L122R^* is required.

## 6. *MYOD1^L122R^* Mutation Can Occur Both in Adults and Children

This section describes the 72 cases of adult and pediatric RMS with somatic mutation of *MYOD1* reported in 10 publications that fulfill our selection criteria, as summarized in Table 1 and Appendix A.

*MYOD1* mutation has been described in both adults and children with RMS, although it appears more frequent in the adult population, particularly in the Sp/Sc subtypes [43]. In the summary of the cases described to date that are shown in Table 1, the majority of patients (43/72, 60%) are ≥18 years of age. Among 29 patients aged <18 years, the median age is 10 years (range 2 to 17 years). Patients aged <10 years represent only 17% (12/72) of all cases described. In pediatric cases, *MYOD1^L122R^* mutations seem to occur in older children and the mutation has not been described in children aged less than 1 year old, as reported previously by Alaggio et al. [13].

According to the observations by Agaram et al. [46], there is no sex preference, with no significant difference in the mutant cohort from the slight majority of affected males seen in pediatric RMS in general. However, among all the published cases, 30/72 (42%) patients are males and 42/72 are females (58%) (Table 1). Moreover, within the pediatric subgroup, 23/29 patients (78%) are female.

## 7. *MYOD1^L122R^* Is Frequent in Sc/Sp Cell Histology but also Present in a Subset of Embryonal RMS

The *MYOD1^L122R^* mutation was first reported in tumors with Sp/Sc histology, with a frequency ranging from 30 to 67% depending on the case series [25,45,46]. Moreover, different authors have pointed out that ERMS can also bear a *MYOD1^L122R^* mutation, in up to 10% of embryonal RMS, most often in adolescents or young adults, with a slightly higher predilection for the female sex and with tumors localized in the head and neck [22,25,45,46]. A recent genomic classification in a large cohort of pediatric RMS showed that *MYOD1* mutations occur in 3% of fusion-negative patients (where “fusion negative” includes tumors without the canonical *PAX3::FOXO1* or *PAX7::FOXO1* fusion, pathognomonic of ARMS) [18]. Moreover, *MYOD1^L122R^* mutations have never been described in any case of *PAX::FOXO1* fusion-positive RMS. Among cases with confirmed *MYOD1* mutations reported in Table 1 which includes both adults and children, 26/72 (36%) of cases had Sp cell histology, 25/72 (35%) had Sc histology, 12/72 (17%) were ERMS and 9/72 (12%) tumors showed a combination of Sp/Sc areas. However, the retrospective character of the analysis and the absence of any centralized pathological review prevents us from drawing strong conclusions about the real *MYOD1 ^L122R^* distribution among histotypes.

## 8. *MYOD1^L122R^* Mutated Tumors Most Frequently Arise at Head and Neck and Extremity Sites

Shern et al.’s genomic characterization of RMS at diagnosis in children included 641 patients from two separate cohorts in the UK and USA, including 515 fusion-negative RMS and 126 fusion-positive RMS [16]. Among the 17 cases of *MYOD1^L122R^* mutated tumors, 16 were located in the head and neck, (including 9/16 (56%) at parameningeal subsite), and in 1 case at the extremities [18]. Among the whole reported cohort of *MYOD1^L1222R^* cases, noting the limitation of the absence of a radiological centralized review, in 31/72 (43%) the primary site was the head and neck (and just 4/31 cases had parameningeal extension) and extremities in 27/72 (38%). In the pediatric subset, the primary site was in the head and neck in 14/28 tumors (50%) and the extremities in 7/28 (25%). Only 3/14 children with head and neck tumors showed parameningeal extension, similar to the low incidence of 1/3 in the series by Owosho et al. [28]. The information regarding nodal extension and distant metastasis at diagnosis is missing for all the cases.

## 9. *MYOD1^L122R^* Mutation Is Associated with a Poor Outcome

The first published report describing Sp- and ScRMS in pediatric and adult patients [43] included a small cohort of nine patients with *MYOD1^L122R^* mutated RMS. Follow-up of at least 12 months was available for 11 patients (including 8/9 *MYOD1 ^L122R^* mutated patients), and among them 9 had an event (64%). In 7/9 patients who had an event, the tumor was *MYOD1 ^L122R^* mutated and in all the cases the event included a distant relapse. Among the 8 *MYOD1^L122R^* mutated RMS patients with a follow-up > 12 months, 2/8 did not have evidence of disease (NED), 2/8 were alive with disease (AWD) and 4/8 were dead of the disease (DOD) [43]. The authors thus concluded that relapse is a frequent event in the subgroup of Sp- and ScRMS bearing *MYOD1^L122R^* mutation.

Moreover, in the report by Alaggio et al. describing only pediatric patients with Sp cell and ScRMS (*n* = 26), the authors underlined the striking differences between pathology in infants and older children [13]. In the infant setting, Sp and Sc histology had wild-type *MYOD1* but showed typical fusion transcripts involving *VGLL* and *NCOA2* as partners. All infantile patients were alive without event. In contrast, in the cohort of 10 older children with Sp or Sc histology and *MYOD1^L122R^* mutation, 7 had early events, with a median time interval of 1.5 years from diagnosis of whom all were dead of disease, 1 patient was alive with disease, 1 had no evidence of disease with a follow-up of 1 year and 1 case was still on treatment. Details of the events are not available, and there is insufficient information to distinguish relapse from primary refractory disease. A better characterization of the events can be found in the report by Agaram et al. published in 2019 [46]. In this cohort of 30 patients, (15 adults, 15 children), with *MYOD1^L122R^* mutated RMS, follow-up was available for 22 patients. Among 20 patients with at least 12 months of surveillance, 16/20 showed at least one event: in 5 cases the event was a local relapse, in 4 cases a distant relapse and in 7 cases a combined local and distant relapse.

The striking difference between the outcome of *MYOD1* wild-type and *MYOD1^L122R^* mutated tumors can be appreciated in the survival analysis by Shern et al. [18]. *MYOD1^L122R^* mutated tumors have a worse event-free survival (EFS) independently of the risk group of classification (5-year EFS in *MYOD1^L122R^* mutated tumors <10% versus MYOD1 wild-type 75%, *p* < 0.0001), based on the known risk factors used in the Children’s Oncology Group (COG) and EpSSG RMS 2005 protocol.

In the cohort described in Table 1, 37/72 (52%) patients had follow-up of at least 12 months (range 12–134 months), including 18 pediatric patients. Information on events was available only for 33/37 (Table 2). Nine patients (24%) had no event. Events were recorded in 24/33 cases (72%) patients. In 8/24 (33%) the event was a local relapse, in 5/24 (21%) a distant relapse and in 11/24 (46%) a combined local and distant relapse. If we focus just on the 18 pediatric patients with follow-up, information on events was recorded for 15 of them: 3/15 (20%) had no event, 5/15 (33%) had a local relapse, 1/15 (7%) had a distant relapse and 6/15 (40%) had a combined local plus distant relapse. Unfortunately, the time interval from diagnosis to the event was not described. Among these 37 patients, 9/37 (24%) were reported as NED, 12/37 (33%) were AWD and 16/37 (43%) were DOD. NED patients had a median age of 34 years (range 8–44 years), AWD patients 24, 5 years (range 8–76), DOD patients 15 years (range 2–77).

Considering only the 24 patients who had an event (local, distant or combined relapse), 2/24 (8%) were NED, 9/24 (37%) were AWD and 13/24 (54%) were DOD (for further detail, see Table 1).

In the DOD group, 6/16 (37%) *MYOD1* mutations were associated with another concurrent molecular event (mutation or copy number variation) in contrast to the NED group in which just 2 out of 9 patients had a concurrent molecular event (22%). For the DOD group, in four cases the associated molecular event was a *PIK3CA* mutation (in one case with a further event being *MDM2* copy number variation), in one an *NRAS* mutation and in one an *MDM2* copy number variation. In 5/6 reports of concurrent molecular events, the patients were in the pediatric group.

## 10. Discussion

In this review, we highlight the importance of the *MYOD1^L122R^* mutation as a prognostic factor for patients with RMS. Our comprehensive analysis of the cases described in the literature confirms the aggressiveness of this specific subgroup of tumors and clarifies the clinical features of patients affected by *MYOD1^L122R^* mutated tumors. There is a lack of information regarding the events and in particular it is not clear if *MYOD1^L122R^* mutated tumors tend to relapse after initial response, or whether they present as primary refractory disease. Similar to the presence of the *PAX::FOXO1* fusion, the presence of this genetic alteration may need to be considered as a factor in risk stratification. This reflection comes from the evidence of a striking difference in survival between *MYOD1* wild-type (fusion negative) and *MYOD1^L122R^* mutated patients, irrespective of the risk group and age [24]. The poor prognosis associated with the presence of the *MYOD1^L122R^* mutation suggests that early identification soon after the diagnosis of these patients at higher risk could be important, particularly if treatment can be tailored accordingly. Although these findings are based on our analysis of cases reported in the literature, the evaluation of the outcome comes from a summary of reported case series that altogether represent a convenience cohort. This could generate potential bias due to non-random selection of samples, as already explained by authors from COG in the case of *PAX::FOXO* biomarker in RMS, and it explains the need to evaluate the role of *MYOD1^L122R^* prospectively in an unselected cohort [50].

Early detection of *MYOD1^L122R^* mutated tumors is facilitated by routine molecular profiling of newly diagnosed tumors, which is now available in some but not all countries. In addition, particular attention to histological features and immunohistochemistry markers is required to detect any possible suggestive constellation of features that could trigger suspicion of a *MYOD1 ^L122R^* mutation and target a subpopulation for analysis of *MYOD1 ^L122R^* mutation. This latter approach could be important for countries where access to molecular biology techniques is a challenge and there is a real need to identify the RMS sub-population most likely to have somatic mutation of *MYOD1*.

The early detection of these patients drives the question of the most adequate staging and treatment to propose once the mutation has been identified. We know that the prognosis seems to be poor when *MYOD1 ^L122R^* mutation is present, but the analyzed reports lack details regarding the role of *MYOD1* mutation as an independent risk factor, the invasiveness of the tumor (for instance, are the nodes more or less frequently involved when *MYOD1* is mutated?), the administered treatment (especially concerning the quality of the local control, and details of radiotherapy) and the pattern of events (is it mostly relapse or are these tumors primary refractory; what is the pattern of relapse?). Better knowledge of these characteristics would shed light on which aspects of the treatment could be improved in this special subset of patients. These reflections ultimately relate to identifying the most adapted practical therapeutic approach when treating these patients within the currently ongoing EpSSG FaR-RMS trial (NCT04625907) in which *MYOD1* is currently not used as prognostic factor in risk stratification.

A consensus of international working groups has recently agreed that molecular analyses of tumors, that include identification of *MYOD1* mutations, should be performed in RMS clinical trials [51]. This will enable the detailed issues raised relating to age and sex predilection and clinical behavior to be comprehensively addressed. However, in light of the correlations with poor outcomes, Shern et al. suggested incorporating *MYOD1^L122R^*, in addition to mutations of *TP53*, into RMS risk stratification [24]. Additionally, a recent report by COG proposed that upstaging on this basis should lead to the intensification of treatment [52] and COG currently excludes cases with *MYOD1^L122R^* tumors from low-risk studies [24,52].

As treatment intensification may not be effective, new treatment strategies should also be sought. These may be derived from more comprehensive molecular knowledge of tumor biology driven by *MYOD1^L122R^* and the role of multiple mutations that are concomitant with *MYOD1^L122R^*. Together with drug screening and testing of patient-derived and other models representing the *MYOD1^L122R^* in RMS, this could open up new therapeutic approaches for this specific group of patients with poor prognosis. More must also be done to design promising innovative drugs and offer these patients the possibility to be included in clinical trials with them.

## 11. Conclusions

In conclusion, if we consider that the *MYOD1* mutation has a similar adverse prognostic role to the presence of *PAX3/7::FOXO1* fusion, which is prospectively assessed in every patient in the EpSSG FaR-RMS trial, *MYOD1* status should be checked early in the course of the disease, to tailor treatment from the beginning, with the aim of improving the survival of these patients. Given the clinical features of the patients that most often have *MYOD1* mutations, we propose to check for *MYOD1* status in patients over the age of 1 year with a *PAX* fusion-negative Sp- or ScRMS localized in the head and neck or extremity site. Ideally, *MYOD1* status should be checked in every pediatric patient with *PAX* fusion-negative histology, to avoid missing the less frequent cases that may have embryonal histology or are localized to other sites.

We propose to prospectively assess the clinical application of *MYOD1^L122R^* status in the FaR-RMS clinical trial as well as consider stratifying these patients based on adverse biology, as for fusion-positive patients, with careful attention to adequate local therapy. In addition, we are currently undertaking a detailed analysis of a cohort of *MYOD1* mutated RMS tumors to help answer some of the detailed clinical and pathological questions raised in this review.

## Figures and Tables

**Figure 1 cancers-15-01644-f001:**
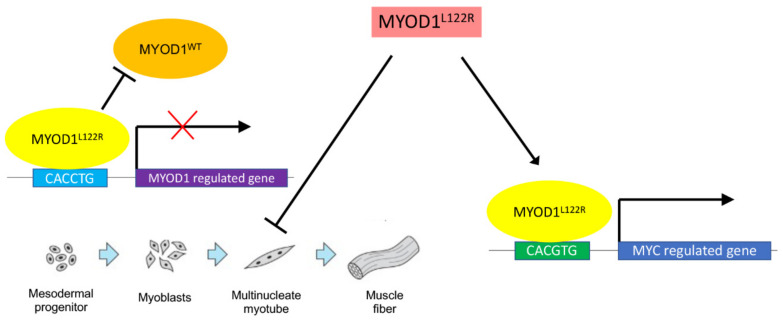
How *MYOD1^L122R^* can contribute towards an aggressive phenotype in RMS. *MYOD1^L122R^* contributes to aggressive disease in RMS, leading to poor outcomes, in two potential ways. Firstly, MYOD1^L122R^ can act in a dominant negative manner and bind MYOD1-responsive promoters, blocking normal MYOD1 from binding but failing to transactivate these myogenic genes, resulting in a failure of myoblasts to differentiate and keeping them in a proliferating state. Secondly, MYOD1^L122R^ can additionally bind to MYC-responsive genes and activate an oncogenic program of transcription.

**Table 1 cancers-15-01644-t001:** Cases of RMS bearing *MYOD1^L122R^* mutation described in literature. Summary of cases of RMS with *MYOD1^L122R^* mutation presented in literature. All patients had *MYOD1^L122R^* mutation. ERMS: embryonal RMS, ScRMS: sclerosing RMS, SCRMS: spindle cell RMS, LR: local relapse, DR: distant relapse, DOD: dead of disease, NED: no evidence of disease, AWD: alive with disease, na: not available. Please note that for other mutations, different methods were used by different authors to assess the co-occurrence of other molecular abnormalities (such as targeted Sanger sequencing, exome sequencing, RNA sequencing).

Study	No	Histological Diagnosis	Age at Diagnosis *(Years)*	Sex	Site	MYOD1 Mutation	Desmin IHC	Myf4 IHC	MyoD1 IHC	Other Mutations	Events	Outcome	Time from Diagnosis to Death, or Last Follow-Up (Months)
Kohsaka 2014 [25]	1	ERMS	21	F	Chest wall	*p*.L122R	na	na	na	no	na	DOD	na
	2	ERMS	37	M	Pterygopalatine fossa	*p*.L122R	na	na	na	PIK3CA	na	DOD	na
	3	ERMS	32	M	Mandible	*p*.L122R	na	na	na	no	na	DOD	na
	4	ERMS	41	F	Facial	*p*.L122R	na	na	na	PTEN deletion	na	na	na
	5	ERMS	32	F	Cheek	*p*.L122R	na	na	na	PTEN deletion	na	na	na
	6	ERMS	4	F	Peritonsillar	*p*.L122R	na	na	na	no	na	na	na
	7	ERMS	35	F	Buccal	*p*.L122R	na	na	na	no	na	DOD	na
	8	ERMS	24	F	Nasopharynx	*p*.L122R	na	na	na	no	na	DOD	na
	9	ERMS	13	F	Hemidiaphragm	*p*.L122R	na	na	na	PIK3CA	na	DOD	na
	10	ERMS	12	F	Infra-temporal fossa	*p*.L122R	na	na	na	PIK3CA	na	DOD	na
Agaram 2014 [43]	11	ScRMS	34	F	Maxilla	*p*.L122R	na	na	na	PIK3CA	DR	NED	41
	12	ScRMS	39	M	Extremities	*p*.L122R	na	na	na	no	na	NED	12
	13	ScRMS	76	M	Extremities	*p*.L122R	na	na	na	no	DR	AWD	17
	14	ScRMS	13	F	Chest wall	*p*.L122R	na	na	na	PIK3CA	LR, DR	DOD	21
	15	ScRMS	14	F	Infra-temporal fossa	*p*.L122R	na	na	na	PIK3CA	LR, DR	DOD	26
	16	SpRMS	10	F	Paraspinal	*p*.L122R	na	na	na	na	LR, DR	DOD	35
	17	SpRMS	2	F	Buttock	*p*.L122R	na	na	na	na	DR	DOD	12
	18	SpRMS	21	M	Pelvis	*p*.L122R	na	na	na	na	LR, DR	AWD	30
	19	SpRMS	35	M	Extremities	*p*.L122R	na	na	na	na	na	AWD	4
Szuhai 2014 [17]	20	SpRMS	52	M	Extremities	*p*.L122R	pos	pos	na	na	na	na	na
	21	SpRMS	32	F	Extremities	*p*.L122R	pos	neg	na	na	na	na	na
	22	SpRMS	28	M	Extremities	*p*.L122R	neg	neg	na	na	na	na	na
	23	SpRMS	71	F	Extremities	*p*.L122R	pos	pos	na	na	na	na	na
	24	SpRMS	24	M	Pharynx	*p*.L122R	pos	pos	na	na	na	na	na
	25	SpRMS	20	F	Mouth	*p*.L122R	pos	pos	na	na	na	na	na
	26	SpRMS	64	M	Extremities	*p*.L122R	pos	pos	na	na	na	na	na
Rekhi 2016 [44]	27	ScRMS	17	M	Extremities	*p*.L122R	na	na	na	na	na	na	na
	28	ScRMS	5	F	Maxilla	*p*.L122R	na	na	na	na	na	AWD	22
Alaggio 2016 [13]	29	ScRMS	17	M	Paraspinal	*p*.L122R	na	na	na	no	na	DOD	24
	30	ScRMS	10	F	Buttock	*p*.L122R	na	na	na	PIK3CA, FGFR4	na	DOD	6
	31	ScRMS	8	M	Extremities	*p*.L122R	na	na	na	no	na	NED	12
	32	ScRMS	11	F	Head	*p*.L122R	na	na	na	PIK3CA	no	on therapy	recent
	33	SpRMS	9	M	Head	*p*.L122R	na	na	na	no	LR	AWD	36
	34	SpRMS	9	F	Head	*p*.L122R	na	na	na	no	no	DOD	12
Owosho 2016 [45]	35	Sp/ScRMS	33	M	Mandible	*p*.L122R	pos	pos	na	no	LR, DR	DOD	65
Agaram 2019 [46]	36	ScRMS	4	F	Extremities	*p*.L122R	na	na	na	no	na	na	na
	37	ScRMS	7	F	Abdominal	*p*.L122R	na	na	na	no	LR, DR	DOD	28
	38	ScRMS	9	F	Head and neck	*p*.L122R	na	na	na	PIK3CA	na	na	na
	39	ScRMS	10	F	Head and neck	*p*.L122R	na	na	na	no	LR	NED	48
	40	SpRMS	17	F	Thorax	*p*.L122R	na	na	na	no	LR, DR	DOD	68
	41	SpRMS	21	M	Pelvis	*p*.L122R	na	na	na	no	LR, DR	DOD	42
	42	Sp/ScRMS	21	F	Head and neck	*p*.L122R	na	na	na	PIK3CA	no	NED	30
	43	Sp/ScRMS	21	F	Head and neck	*p*.L122R	na	na	na	na	no	na	na
	44	Sp/ScRMS	26	M	Extremities	*p*.L122R	na	na	na	PIK3CA	no	NED	4
	45	ScRMS	31	F	Head and neck	*p*.L122R	na	na	na	PIK3CA	LR	AWD	12
	46	SpRMS	33	M	Extremities	*p*.L122R	na	na	na	no	na	na	na
	47	SpRMS	36	M	Extremities	*p*.L122R	na	na	na	PIK3CA	LR, DR	DOD	16
	48	SpRMS	38	F	Extremities	*p*.L122R	na	na	na	no	na	na	na
	49	Sp/ScRMS	39	M	Extremities	*p*.L122R	na	na	na	no	no	NED	60
	50	SpRMS	44	F	Paraspinal	*p*.L122R	na	na	na	no	no	NED	13
	51	SpRMS	45	M	Liver	*p*.L122R	na	na	na	no	na	na	na
	52	SpRMS	77	M	Extremities	*p*.L122R	na	na	na	no	DR	DOD	32
	53	ScRMS	94	M	Extremities	*p*.L122R	na	na	na	no	na	na	na
Tsai 2019 [47]	54	Sp/ScRMS	42	F	Head	*p*.L122R	pos	pos	pos	no	no	NED	134
	55	ScRMS	23	F	Parapharynx	*p*.L122R	pos	pos	pos	no	LR, DR	DOD	24
	56	SpRMS	34	F	Mediastinum	*p*.L122R	pos	pos	pos	no	LR	AWD	12
	57	Sp/ScRMS	64	M	Extremities	*p*.L122R	pos	pos	pos	no	LR	AWD	13
	58	ScRMS	22	F	Extremities	*p*.L122R	pos	pos	pos	no	no	NED	2
	59	ScRMS	15	F	Parapharynx	*p*.L122R	pos	pos	pos	no	no	AWD	13
	60	SpRMS	42	M	Extremities	*p*.L122R	pos	pos	pos	no	DR	AWD	12
	61	SpRMS	8	F	Head	*p*.L122R	pos	pos	pos	no	DR	AWD	6
	62	ScRMS	28	M	Head	*p*.L122R	pos	pos	pos	no	no	AWD	14
	63	Sp/ScRMS	8	F	Extremities	*p*.L122R	pos	pos	pos	no	LR	AWD	29
	64	SpRMS	19	F	Pterygomandibular	*p*.L122R	pos	pos	pos	no	no	NED	50
	65	Sp/ScRMS	16	F	Pre-auricular	*p*.L122R	pos	pos	pos	no	LR	AWD	24
Gorunova 2020 [48]	66	ScRMS	3.5	F	Extremities	*p*.L122R	na	na	na	na	na	na	na
	67	ScRMS	30	M	Extremities	*p*.L122R	na	na	na	na	na	na	na
Ting 2021 [49]	68	ERMS	10	M	Retroperitoneum	*p*.L122R	na	na	na	PIK3CA, MDM2 gain	LR	DOD	24
	69	ScRMS	20	M	Chest wall	*p*.L122R	na	na	na	no	LR, DR	DOD	10
	70	ScRMS	15	F	Extremities	*p*.L122R	na	na	na	NRAS	LR, DR	DOD	49
	71	SpRMS	15	M	Head and neck	*p*.L122R	na	na	na	MDM2 gain	no	DOD	13
	72	ERMS	13	F	Head and neck	*p*.L122R		na	na	no	na	AWD	na

**Table 2 cancers-15-01644-t002:** Events and outcomes in children and adults. Summary of cases with a follow-up of at least 12 months with events and outcome, by age. na: not available, LR: local relapse, DR: distant relapse, NED: no evidence of disease, AWD: alive with disease, DOD: dead of the disease.

	Children (*n* = 18)	Adults (*n* = 19)	Tot (*n* = 37)
Events			
na	3	1	4
No	3	6	9
Yes	12	12	24
* Type of event *			
* LR *	*5*	*3*	*8*
* DR *	*1*	*4*	*5*
* LR + DR *	*6*	*5*	*11*
Outcome			
NED	2	7	9
AWD	5	7	12
DOD	11	5	16

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
