# Peer review of "Biological Role and Clinical Implications of MYOD1L122R Mutation in Rhabdomyosarcoma"

_cancers, 2023, doi:10.3390/cancers15061644_

Round 1

Reviewer 1 Report

This work proposes an extensive review of the biological role and clinical implications of MYOD1 mutation in Rhabdomyosarcoma.  I have several comments for the authors’ consideration to further improve the manuscript:

1)    There are some minor language errors. The authors should be revised the manuscript with an English language editor to make it more readable.

2) The author mentioned that “MYOD1L122R mutation can occur both in adults and children” and the data that they used to support their results showed that the majority of patients were adults. However, rhabdomyosarcoma was more commonly seen in children, I suggest the author further discuss these differences in the discussion part.

3)   Some literature cited was too old. The most updated information in the research field should be mentioned in the manuscript.

Author Response

REVIEWER 1

Mutation This work proposes an extensive review of the biological role and clinical implications of MYOD1 mutation in Rhabdomyosarcoma.  I have several comments for the authors’ consideration to further improve the manuscript:

1) There are some minor language errors. The authors should be revised the manuscript with an English language editor to make it more readable.

Thank you for this comment. Our English co-authors have thoroughly revised the manuscript to ensure that the article reads well.

2) The author mentioned that “MYOD1L122R mutation can occur both in adults and children” and the data that they used to support their results showed that the majority of patients were adults. However, rhabdomyosarcoma was more commonly seen in children, I suggest the author further discuss these differences in the discussion part.

Thank you for this comment. The adults/paediatric ratio could also be the result of a bias of mutation research and selective reporting. It is a single finding that does not mirror necessarily MYOD1 epidemiology. We have added a comment in the discussion that prospective testing will help us to understand age and sex predilection.

3)   Some literature cited was too old. The most updated information in the research field should be mentioned in the manuscript.

Thank you for this comment. The concept at the basis of this work was to offer a comprehensive review of MYOD1 mutation (focused on RMS) from the first time it was described up to now. The most relevant recent papers have been all cited, but we think there is value in keeping the older ones to describe how the MyoD1 story has developed.

Reviewer 2 Report

Daniela et al, did an elaborate review on role of MYOD1L122R mutation in Rhabdomyosarcoma. This article is very well written. 

Author Response

Thank you very much for this kind comment.

Reviewer 3 Report

This is a comprehensive literature review of studies on rhabdomyosarcoma cases with the MYOD1 L112R mutation by Carlo et al.  Overall, the review has provided informative findings from the past studies on clinical, molecular and biological characteristics of these tumors. I recommend acceptance following minor revision based on my comments below:

1. Line 111: The authors mentioned that contributions in English, French and Italian were considered.  What is the rationale for this inclusion?

2. The authors included the percentages of MYOD1 L112R-mutated cases based on various parameters.  It was at times unclear at times whether they were referring to pediatric, adult or mixed populations.  For example, do the percentages listed for histologic subtypes (lines 255-257) and anatomic locations (lines 267-270) include both pediatric and adult cases?

3. The review included findings of pediatric and adult MYOD1 L112R-mutated cases.  It might be a good idea to have a separate subsection summarizing similarities and differences in these two populations.

4. Line 324: What concurrent molecular events with the MYOD1 L112R cases were assessed besides PIK3CA, NRAS and MDM2?

5. The discussion section had mentioned MYOD1 L112R being somatic (line 354). Has the germline status of MYOD1 L112R ever been assessed?

6. Lines 320-321: What study did the 24 patients referred to here come from? How is “an event” defined here?

7. Other minor comments:

            - Some abbreviations need to be spelled out first e.g. PRIMA (line 48) and EFS (line 302).

            - In the abstract, “embryonal tumor” in line 51 might be better referred to as embryonal subtype.

            - In table 1, instead of the label “paper” in the first column, an alternative option is “study”.

Author Response

REVIEWER 3

This is a comprehensive literature review of studies on rhabdomyosarcoma cases with the MYOD1 L112R mutation by Carlo et al.  Overall, the review has provided informative findings from the past studies on clinical, molecular and biological characteristics of these tumors. I recommend acceptance following minor revision based on my comments below:

  1. Line 111: The authors mentioned that contributions in English, French and Italian were considered.  What is the rationale for this inclusion?

The decision to include papers not in French and Italian as well as English was based on the languages known by the authors’ team. We are not aware of significant papers in other languages that are relevant to this review.

  1. The authors included the percentages of MYOD1 L112R-mutated cases based on various parameters.  It was at times unclear at times whether they were referring to pediatric, adult or mixed populations.  For example, do the percentages listed for histologic subtypes (lines 255-257) and anatomic locations (lines 267-270) include both pediatric and adult cases?

Thank you for this comment. If not otherwise specified, the percentages refer to the whole population, but we have added a specific comment to make it clear at the start of section 6 and in section 7 that Table 1 includes both adults and children.

  1. The review included findings of pediatric and adult MYOD1 L112R-mutated cases.  It might be a good idea to have a separate subsection summarizing similarities and differences in these two populations.

Thank you for this suggestion. We have included a new Table (Table 2) showing outcomes in paediatric patients, adult patients and the whole group.  

  1. Line 324: What concurrent molecular events with the MYOD1 L112R cases were assessed besides PIK3CA, NRAS and MDM2?

Different methods were used by different authors so assessment of molecular events was not uniform. Therefore we have not addressed this specifically in the manuscript.

e.g.:

  • Kohsaka 2014 RNA seq,
  • Agaram 2014 targeted Sanger sequencing for MYOD and PIK3CA,
  • Alaggio 2016 RNA seq for 4 cases, but also targeted PCR and Sanger sequencing for MYOD1, PIK3CA
  • Agaram 2019 targeted PCR and Sanger sequencing for MYOD and PIK3CA plus MSK-impact assay (screening for > 400 oncogenes and onco suppressors)
  • Ting 2021 targeted Sanger sequencing for MYOD for some cases, exome sequencing for other cases

  1. The discussion section had mentioned MYOD1 L112R being somatic (line 354). Has the germline status of MYOD1 L112R ever been assessed?

We are not aware of reports of germline MYOD1 status check.

  1. Lines 320-321: What study did the 24 patients referred to here come from? How is “an event” defined here?

The 24 patients come from different studies. Table 1 identifies them by report. We have added the comment (for further detail see Table 1) to make it clear. “Event” is defined based on the definition used in the different studies as local, distant or combined relapse. It is not clear whether there are also patients with primary refractory tumors. We discuss this issue in the discussion section.

  1. Other minor comments:

            - Some abbreviations need to be spelled out first e.g. PRIMA (line 48) and EFS (line 302).

            - In the abstract, “embryonal tumor” in line 51 might be better referred to as embryonal subtype

            - In table 1, instead of the label “paper” in the first column, an alternative option is “study”.

Thank you, these comments have been addressed.
